# EduGym: An Environment Suite for Reinforcement Learning Education

## Abstract

Due to the empirical success of reinforcement learning, an increasing number of students study the subject. However, from our practical teaching experience, we see students entering the field (bachelor, master and early PhD) often struggle. On the one hand, textbooks and (online) lectures provide the fundamentals, but students find it hard to translate between equations and code. On the other hand, public codebases do provide practical examples, but the implemented algorithms tend to be complex, and the underlying test environments contain multiple reinforcement learning challenges at once. Although this is realistic from a research perspective, it often hinders educational conceptual understanding. To solve this issue we introduce *EduGym*, as a set of *educational reinforcement learning environments* and associated *interactive notebooks* tailored for education. Each EduGym environment is specifically designed to illustrate a certain aspect/challenge of reinforcement learning (e.g., exploration, partial observability, stochasticity, etc.), while the associated interactive notebook explains the challenge and its possible solution approaches, connecting equations and code in a single document. An evaluation among RL students and researchers shows 86% of them think EduGym is a useful tool for reinforcement learning education. All notebooks are available from this link,[1] while the full software package can be installed from `www.github.com/anonymized`.

## 1 Introduction

Reinforcement learning has become an important area of artificial intelligence research, for example showing strong empirical performance in algorithm design (Fawzi et al., 2022), physical process control (Degrave et al., 2022), drug design (Popova et al., 2018), molecular design (Zhou et al., 2019), optimization of chemical reactions (Zhou et al., 2017), natural language processing (OpenAI, 2023), and game playing (Silver et al., 2018). Following this success, an increasing number of students are entering the field, either at bachelor, master or early PhD level (depending on your study programme). These students of course need teaching material, for which textbooks (Sutton & Barto, 2018; Wiering & Van Otterlo, 2012), online lecture series (Silver, 2015; van Hasselt et al., 2021; White & White, 2023) and documented codebases (Achiam, 2018; Raffin et al., 2019) are most popular.

Despite this large offer of reinforcement learning teaching material, we still see students struggle when entering the field. From our practical teaching experience, we identify a few underlying causes. First of all, textbooks probably provide the most complete introduction to all fundamental concepts. Reinforcement learning is a modular field: we may combine different exploration methods (e.g., $\epsilon$-greedy) with different credit assignment methods (e.g., a one-step on-policy estimate), different partial observability methods (e.g., windowing/frame-stacking), etc. Textbooks excel at independently discussing these topics, with conceptual examples. However, we often see students struggle to translate equations into actual code (and vice versa). Without an interactive practical example, we see many students get lost or their learning cycle takes much longer.

A potential solution to this problem is a documented public codebase, which allows for active experimentation by the student. However, here they run into a different kind of problem since most

---

[1] `https://sites.google.com/view/edu-gym/home`. Anonymised link that will be replaced with a proper URL for a final paper version.

Table 1: Reinforcement learning challenges and associated environments included in EduGym.

| | Challenge | Environment | Variable parameters |
|---|---|---|---|
| a | Exploration | Boulder | Height, width |
| b | Credit assignment: On/off-policy | Roadrunner | Negative event reward size |
| c | Credit assignment: Depth | Study | # irrelevant actions, reward noise |
| d | State: Dimensionality | Catch | Observation type, problem size |
| e | State: Partial observability | MemoryCorridor | |
| f | State: Amount of signal | Tamagotchi | Message noise, vocabulary size |
| g | State/action: Discrete – continuous | Trashbot | Discretization bin size |
| h | Dynamics: Stochasticity | Golf | Stochasticity level |
| i | Model-based reinforcement learning | Supermarket | Step timeout, model noise |

of these codebases are research-oriented. They implement state-of-the-art algorithms that combine many ideas (a complicated exploration method, a neural network for representation learning, etc.), making it hard for students to disentangle the separate challenges and their possible solutions. In addition, the codebases usually contain much boilerplate code (for logging, parallelisation, etc.), with the actual 'reinforcement learning magic' deeply hidden away.

To further complicate matters, most reinforcement learning test environments (nearly all based on the Gym paradigm (Brockman et al., 2016)) are also complicated. Most of these environments are high-dimensional, which makes the experimentation cycle slow. Representation learning is of course an important challenge, but may actually be a hindrance when we want to teach a student about the difference between on- and off-policy back-ups. For low-dimensional (teaching) experiments educators often resort to classic control tasks such as *CartPole* and *MountainCar*, but these have very particular characteristics (also confounding multiple challenges) that may not be suitable to illustrate each aspect of reinforcement learning algorithms.

In short, there appears to be a gap in reinforcement learning education. We first of all require specific educational reinforcement learning environments, that are 1) designed to isolate a particular aspect/challenge of reinforcement learning, 2) have tunable parameters to vary the strength of that particular challenge, and 3) allow for a fast experimentation cycle. In addition, we need a set of interactive notebooks, that 1) illustrate the particular challenge and its possible solution approaches, connecting equations and code in a single document, and 2) allow for interactive experimentation by the student, which is known to be a key requirement for fast learning progress (Abdel Meguid & Collins, 2017). EduGym provides exactly this contribution:

- We provide a set of *educational reinforcement learning environments*, each specifically designed for a particular reinforcement learning challenge or design decision.

- We provide a set of *interactive notebooks*, where students are gradually taught about each challenge, and can at the same time actively experiment with relevant code.

As such, EduGym is intended as an interactive companion to other reinforcement learning teaching material. The challenges that we included, and the associated environments we designed for each, are listed in Table 1. The full EduGym package is available from `www.github.com/anonymized`, while all interactive notebooks are accessible through `https://sites.google.com/view/edu-gym/home`.

## 2 RELATED WORK

**Environment suites** *Gymnasium* (Brockman et al., 2016; Towers et al., 2023) has been the most popular framework for standardization of reinforcement learning environments. The core of Gymnasium contains well-known low-dimensional problems, such as *CartPole*, *MountainCar* and *FrozenLake*, as well as a range of more complicated *Mujoco* (Todorov et al., 2012) and *Atari 2600* (Bellemare et al., 2013) environments. The Gym framework has sparked the development of a variety of follow-up environment suites, on a wide range of topics. Examples include Gym environments for multi-agent reinforcement learning (Terry et al., 2021), multi-objective reinforcement learning (Alegre et al.,

2022), meta reinforcement learning (Yu et al., 2019), object-centric learning (Watters et al., 2019), safety (Ray et al., 2019), visual navigation (Kempka et al., 2016; Tunyasuvunakool et al., 2020; Szot et al., 2021), procedural content generation (Cobbe et al., 2019), robotics (OpenAI, 2020; de Lazcano et al., 2023), traffic control (Wu et al., 2017), energy management (Findeis et al., 2022), etc. While all these environments are well-suited for research, they are usually not ideal for education. Specifically, and in contrast to EduGym, they are usually 1) high-dimensional (leading to a slow experimentation loop), 2) combine several types of challenges (making it hard for students to disentangle them), and 3) do not have modifiable parameters (making it hard for students to systematically scale the intensity of a particular challenge). An exception is *MemoryGym* (Pleines et al., 2023), which does specifically focus on partial observability, but is research-oriented (relatively challenging tasks) and comes without explanatory notebooks.

**Toy environments**    EduGym instead contains a range of toy environments specifically designed for a particular challenge of reinforcement learning (Table 1). For some of these challenges there are other toy examples in literature. For example, a classic toy task for the distinction between on-policy and off-policy updates is the *Cliff Walking* environment from Sutton & Barto (2018). For (sparse reward) exploration, there are *The Chain* (Strens, 2000) and *Deep Sea* (Osband et al., 2020), while for partial observability there is the *Tiger* problem (Kaelbling et al., 1998). For EduGym we nevertheless designed a set of novel environments, for several reasons. First of all, not all included challenges had clear available toy examples, such as stochasticity, state signal and model-based RL. Second, the available toy examples usually lack parameters to tune the difficulty of the challenge. Finally, we believe it is educationally beneficial for students to be able to recognize the same challenge in different settings. Existing Gymnasium toy environments can easily be swapped into our notebooks, which we for example show for the Cliff Walking environment in the notebook on on-policy versus off-policy learning.

**Algorithmic frameworks**    Several researchers have developed software frameworks for reinforcement learning research. These usually either provide standardised implementations of popular algorithms and/or provide a modular framework to allow for a fast experimentation loop. Example frameworks include *RLLib* (Liang et al., 2018), *Dopamine* (Castro et al., 2018), *MushroomRL* (D'Eramo et al., 2021), and *ChainerRL* (Fujita et al., 2021). All these frameworks have their own flavour, but on an abstract level they are all *research* oriented: they implement relatively complicated algorithms, and the codebase contains much boilerplate code for logging, computational efficiency, etc. Some frameworks, notably *Stable Baselines* (Raffin et al., 2019) and *SpinningUp* (Achiam, 2018), do come with additional educational explanation of the implemented algorithms. However, they still discuss relatively complex algorithms with a range of tricks implemented in highly modularized codebases. Instead, EduGym focuses on students who have not yet reached this level, and desire a codebase focused on simplicity, insight and a quick experimentation loop.

**Benchmarking**    EduGym covers a wide range of challenges, and therefore also has a relation to benchmarking. While any task may in principle serve as a benchmark, we prefer a benchmark set that covers all relevant challenges. In recent years, there has been a lot of emphasis on one particular challenge (state dimensionality), which is reflected in the many high-dimensional Gym environments mentioned above. However, Osband et al. (2020) correctly emphasized that RL problems can be challenging in different ways, of which state dimensionality is only one aspect. The authors introduce *Behavioural Suite for Reinforcement Learning*, which benchmarks the strength of a particular algorithm in different environments with a particular challenge. EduGym takes the same modular approach, but our focus instead resides on education. The main difference is, of course, our set of educational notebooks, while we also include more challenges/environments, and design them with adjustable parameters. However, Behavioural Suite was an important inspiration, while EduGym itself could vice versa also be used as a benchmark suite.

## 3    BACKGROUND

We adopt a Markov Decision Process (MDP) formulation defined by the tuple $(\mathcal{S}, \mathcal{A}, p, r, \gamma)$, where $\mathcal{S}$ is a set of states, $\mathcal{A}$ is a set of actions, $p : \mathcal{S} \times \mathcal{A} \to P(\mathcal{S})$ is a transition function, $r : \mathcal{S} \times \mathcal{A} \times \mathcal{S} \to \mathbb{R}$ is a reward function, and $\gamma \in [0, 1]$ is a discount factor. At each timestep $t$, the agent observes a state $s_t \in \mathcal{S}$ and chooses an action $a_t \in \mathcal{A}$, after which it observes a next state $s_{t+1} \sim p(s_{t+1}|s_t, a_t)$

and associated reward $r_t = r(s_t, a_t, s_{t+1})$. The agent selects these actions based on a policy $\pi : \mathcal{S} \to P(\mathcal{A})$, which specifies the probability of each action given a particular state.

Our goal is to find the policy $\pi(a|s)$ that obtains the best average sum of discounted rewards. Define the state-action value function $Q^\pi(s, a)$ as the expected cumulative reward after taking action $a$ in state $s$ and following policy $\pi$ afterwards:

$$Q^\pi(s, a) = \mathbb{E}_{\pi,p}\left[\sum_{i=0}^{\infty}(\gamma)^i \cdot r_{t+i}\Big|s_t = s, a_t = a\right] \tag{1}$$

We then want to find the optimal value function $Q^\star(s, a) = \max_\pi Q^\pi(s, a)$ and (one of) its associated optimal policies $\pi^\star(a|s) = \arg\max_\pi Q^\pi(s, a)$. In reinforcement learning, we attempt to learn this optimal policy or value function through trial and error, where we interact with the environment to gradually improve our solution estimates.

## 4 CHALLENGES AND ENVIRONMENTS

We will first discuss the challenges considered in EduGym and the associated environment designed to illustrate each challenge. An overview of all challenges and associated environments is presented in Table 1, which also lists the parameters that may be varied per environment (which usually allows for scaling the strength of a particular challenge). Full technical details of each environment are provided in Appendix A.

**a. Exploration: Boulder**    A first fundamental reinforcement learning topic is the *exploration-exploitation* trade-off: we sometimes need to try novel action(s) (sequences), but we also want to commit to what works well to make progress in the task (Amin et al., 2021). This challenge, which lies at the heart of reinforcement learning, is especially pronounced in the case of *sparse* rewards, where most transitions have the same reward and interesting different rewards are hidden away in specific deep action sequences. This makes them hard to discover for standard exploration methods based on random perturbation (Osband et al., 2016). To illustrate the challenge of (sparse) exploration we designed the *Boulder* environment (inspired by the *Chain* task of Strens (2000)). The agent has to climb a boulder with a fixed number of grips at every step towards the top. However, at every height, only one of the available grips provides enough hold, while the other actions make the agent fall back down. Therefore, the agent has to repeatedly explore the boulder, like a (secured) rock climber, to find out which path actually gets it to the top. The task has a sparse reward at the top, with all other transitions providing zero reward. Due to this sparse reward, random exploration methods suffer, since they essentially have to luckily sample the exact correct sequence of actions to ever see a non-zero reward. The exploration challenge of this task can be modified by adjusting the height and width of the Boulder at initialization.

**b. On/off-policy: Roadrunner**    Another fundamental topic in reinforcement learning is the way we back-up reward information, i.e., how we compute new estimates of the expected cumulative reward for a certain state-action pair. A crucial distinction in the back-up is the used policy, for which we discriminate on-policy (Rummery & Niranjan, 1994) and off-policy (Watkins & Dayan, 1992) methods. *On-policy* updates compute the value of the current behavioural policy, taking into account potential exploration, while *off-policy* back-up the value of the best available next action, updating as if we will greedily exploit eventually (switching off exploration). To illustrate this difference, we designed the *Roadrunner* environment, inspired by the classic *Looney Tunes* cartoon where *Wiley E. Coyote* chases *Road Runner* towards a cliff (with Road Runner stopping just in time, while Wiley E. Coyote overshoots and falls over the edge). Our agent has to speed up towards a cliff as quickly as possible, getting rewarded for both the speed with which it reaches the cliff, and how close it ends up to the edge. The intuition of this task is that on-policy methods will obtain a conservative solution, as they are learning the value function of the exploratory policy. An exploratory step at the cliff edge may make us fall in, so stopping a little distance from the cliff is preferred. Instead, the off-policy method will learn the optimal solution (under the greedy policy), and thereby stop just at the cliff edge.

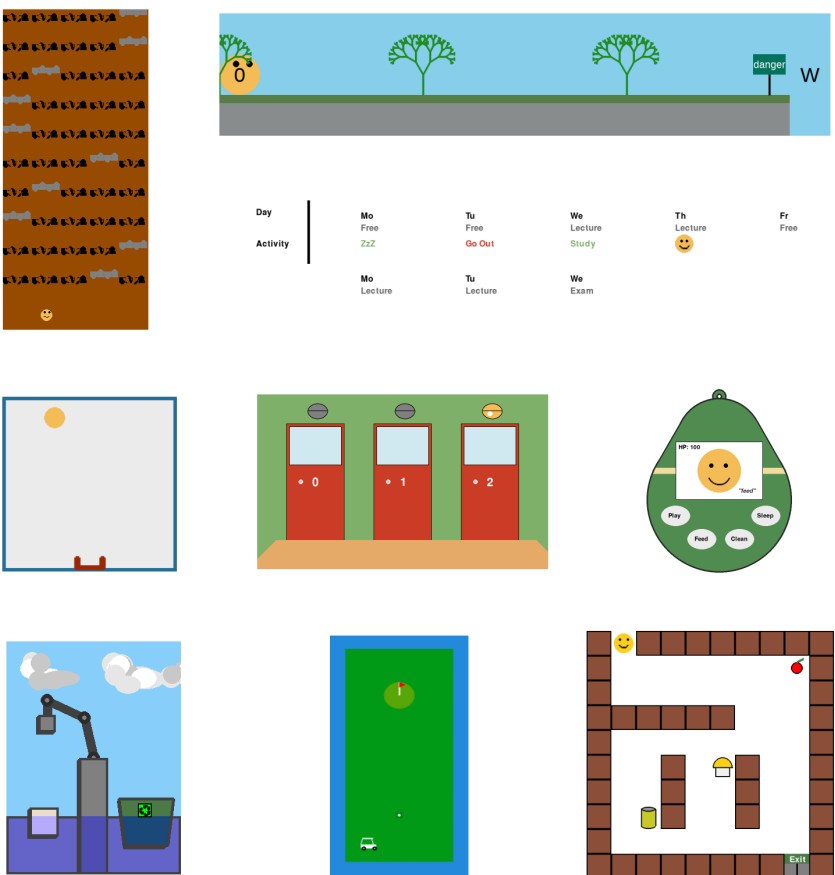

Figure 1: Edugym environments. Top row: Boulder (illustrates exploration), Roadrunner (illustrates on/off-policy), Study (illustrates credit assignment depth). Middle row: Catch (illustrates dimensionality), MemoryCorridor (illustrates partial observability), Tamagotchi (illustrates state signal). Bottom row: Trashbot (illustrates continuous states/actions), Golf (illustrates stochasticity), Supermarket (illustrates model-based reinforcement learning). All environments are introduced in section 4, while full specifications are available in Appendix A.

**c. Credit assignment depth: Study**    A second key consideration regarding the back-up phase is its *depth*. On the one hand, 1-step back-ups give low variance updates (and easily allow for off-policy updating), but 1-step back-up estimates are biased and propagate information relatively slowly. On the other extreme, Monte Carlo back-ups give unbiased estimates and allow for fast propagation of information, but they suffer from high variance. This leads to the well-known *bias-variance* trade-off in reinforcement learning (Sutton & Barto, 2018). To illustrate this trade-off we designed the *Study* environment. In this task, the agent is a student who has to decide per day what they will do: sleep in, study, or go out. On certain days there are lectures at the university, and the agent needs to study on those days to increase its knowledge. In addition, the chosen action per day also affects the energy of the agent. On the last day of the semester (of adjustable length), the agent takes the exam, which it passes if both its knowledge and energy are high enough. The intuition of this task is that the agent has some crucial decisions (study on the days when there are lectures), while the other days are less relevant and simply introduce noise into the return signal. The Monte Carlo agent, in this case, learns to solve the task fairly quickly after the first successful episode due to the fast propagation of information, but then has trouble eliminating the suboptimal actions along the path. On the contrary, one-step methods learn more slowly initially but are later on better able to filter out suboptimal actions. Students can increase the variance in the returns (by increasing the reward noise and/or adding irrelevant actions with random reward), which will make Monte Carlo methods suffer more.

**d. State dimensionality: Catch**   One of the dominant aspects in all machine learning is the (input) *dimensionality* of the problem. When the dimension of your input and/or output grows, you can no longer store the solution in a table, let alone observe all possible states. We then require *function approximation* to learn informative representations of the input and benefit from generalisation to achieve a good solution for the entire task (Bengio et al., 2013). To illustrate these considerations, we use the *Catch* environment (inspired by Osband et al. (2020)). In Catch, the agent controls a padel at the bottom of the screen, which needs to catch balls dropping from the top of the screen. The main feature of this task is that we can choose the type of observation: a vector-based compact representation, consisting of the padel $x$ and ball $x$ and $y$ locations, or a pixel-based high-dimensional representation of the full screen. The vector representation is of course fully informative and allows for a quick tabular solution. However, in more complicated tasks good representations are usually not known, and we have to rely on the high-dimensional raw input. In those cases, function approximation may still solve the pixel-based Catch variant. Students can adjust the type of observation and the scale of the problem (width and height).

**e. Partial observability: MemoryCorridor**   Another key consideration in reinforcement learning is the *observability* of the task. Many real-world tasks are *partially observable* (Jaakkola et al., 1994), where the current observation does not provide all information about the true state of the MDP. In these cases, information from previous timesteps ('memory') may help to better identify the true state, and therefore find a good policy. To illustrate partial observability, we designed the *MemoryCorridor*, in which the agent needs to navigate a corridor of doors. At each step in the corridor, the agent sees three available doors. However, only one of the doors is truly unlocked (which brings the agent to the next step), while all other doors are locked and terminate the episode without reward. During an episode, the agent enters a sequence of corridors that gradually increase in length: if it solves the corridor of length 1, then it will next face a corridor of length 2, etc. The sequence of correct actions throughout each corridor stays the same within an episode, but the correct, unlocked door is only marked at the last step. The agent therefore needs to remember the sequence of correct doors to reach the end of the corridor, where the correct next door is marked again. Because the length of the faced corridors keeps increasing when the agent does well, this task automatically scales its memory challenge.

**f. Amount of state signal: Tamagotchi**   We often see students throw a machine learning method at a dataset/problem that clearly does not contain enough signal to solve the task. Clearly, machine learning is not a magic tool; if you could never solve the available task with the available information yourself, then an agent probably will not be able to do so either. One way in which information can come to the agent in compact form is through natural language, which has shown to be a powerful representation space (OpenAI, 2023) and will therefore be used for illustration. To demonstrate the importance of state signal (and language), we designed the *Tamagotchi* environment. The agent needs to control a classic Tamagotchi toy based on four available actions: play, feed, sleep and clean. Each of these actions affects an associated internal variable of the Tamagotchi, which together determine the obtained reward. However, the agent cannot directly observe these internal variables but only sees its hitpoints ('HP') derived from all internal variables together. The agent does not know which internal variable is currently deficient and what action needs to be chosen. However, the Tamagotchi does generate utterances related to the current internal variables and may thereby provide the necessary state information. Students can vary the signal-to-noise ratio in the Tamagotchi's communication (observing the effect of state information), and adjust the vocabulary size and utterance length (studying the language component).

**g. Discrete/continuous tasks: Trashbot**   Another core issue in reinforcement learning is the distinction between *discrete* and *continuous* state and action spaces. Most tasks in EduGym use discrete spaces because their solutions are more easy to track. However, many real-world reinforcement learning tasks, especially in the field of robotics (Kober et al., 2013), have continuous observations (e.g., joint angles) and actions (e.g., torques on motors). For continuous *state* spaces, we may either use *discretisation* for a tabular solution, or use *function approximation*, which naturally handles continuous inputs. For continuous *action* spaces, we may again use *discretisation*, which may lose some fine-grained accuracy, or a *policy-based* reinforcement learning method (Deisenroth et al., 2013), which directly learns a distribution over the continuous output space. To illustrate the effect of continuous state and action spaces we designed the *Trashbot* environment. The agent controls a robot

that needs to pick up trash and drop it into a garbage container. The robot is rewarded for the precision with which it drops elements into the container: the closer to the middle it drops the trash, the higher the reward. Students can instantiate Trashbot with either a continuous or a discretised action space, where one can also adjust the number of bins in the discretised setting. They can then investigate how continuous solution methods allow for a more fine-grained solution, and may therefore converge to a higher expected return in this task, while discretization may trade-off a decrease in accuracy with an increase in learning speed (an effect that also varies with the number of discretisation bins).

**h. Stochasticity: Golf**   Students also need to learn about the influence of *stochastic* dynamics on the performance of reinforcement learning agents. First of all, noise in the transition function generally makes learning more challenging, because we need more samples to estimate the expected return of states and actions. In addition, stochasticity also plays a role in *risk-sensitive reinforcement learning* (Neuneier & Mihatsch, 1998; Garcıa & Fernández, 2015). In these cases, which for example occur in financial portfolio management, we do not only care about the best average return, but also want to avoid large return fluctuations. We therefore may want to penalize actions that lead to high stochasticity, even when they do good on average. To illustrate the effect of stochasticity, we designed the *Golf* environment. The agent faces a golf course where it needs to hit the ball into the hole. For each hit the agent can pick from three actions: 'put', 'chip' and 'drive' (a short, medium and hard hit). Every hit moves the ball in the direction of the hole, but the endpoint of the ball has a stochastic component: we add independent Gaussian noise as a displacement of the ball, representing the skill level of the player. The key point of the environment is that the amount of noise increases with the hitting strength: the harder we hit (e.g., 'drive'), the stronger the ball may deviate. Students can vary the skill level of the player (i.e., the amount of stochasticity in the dynamics) and observe the effect on learning speed, and can also try out risk-sensitive reinforcement learning methods, which lead to a more risk-averse or risk-seeking style of play.

**i. Model-based reinforcement learning: Supermarket**   The final topic we included in EduGym is *model-based reinforcement learning*. Interaction with a real-world environment is usually costly: it takes time and energy to execute actions, and the consequence of a wrong exploratory action might be severe. Humans instead have the ability to *plan* in their mind: predicting the effect of certain sequences of actions with an internal *model* of the environment. This idea is the basis of model-based reinforcement learning, where the agent plans over a (learned) model to make additional updates to the solution (data efficiency), decide on better actions, guide exploration, etc. (Moerland et al., 2023) To illustrate the effect of model-based reinforcement learning, we designed the *Supermarket* environment. Here, the agent needs to navigate through a supermarket to collect three items on its shopping list (each for a positive reward), to eventually leave the supermarket through the exit (for an additional terminal reward). The Supermarket has two special properties that make it specifically suited for model-based reinforcement learning. First of all, students can set a *timeout* parameter that determines how long the environment step function will block after each call. This simulates the cost of real-world actions, which proceed much slower than simulated ones. During the timeout, the agent cannot take any action in the environment, but it can plan with an internal model, thereby effectively setting the planning budget per timestep. Second, the Supermarket also comes with a readily available *model function*, which can either generate full next-state probabilities (descriptive model) or next-state samples (generative model). With this model, students can choose to skip the model learning phase, and directly investigate the effect of planning methods. In addition, they can also vary the amount of random noise in the model (mimicking model uncertainty), and observe its effect on learning performance.

## 5   INTERACTIVE NOTEBOOKS

The second key part of EduGym are the *interactive notebooks*, which each illustrate one of the challenges listed in Table 1. Each notebook introduces the particular environment, explains its challenge, covers possible solution approaches, and experimentally illustrates its performance. All notebooks are available from `https://sites.google.com/view/edu-gym/home`. Here, we provide a few illustrative examples of results in these notebooks.

**Exploration in the Boulder environment**   The interactive notebook on exploration, which uses the Boulder environment, includes Figure 2. Here, we study the performance of different exploration

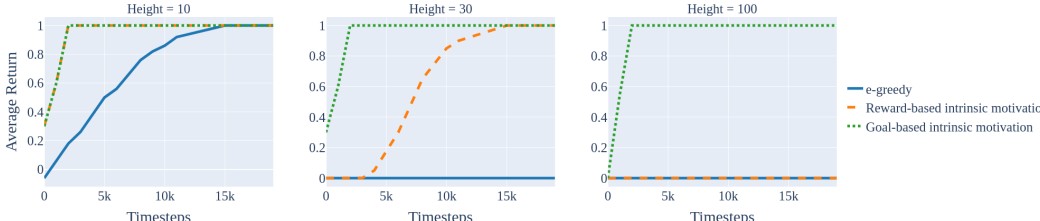

Figure 2: Comparison of different exploration methods on the *Boulder* environment. Subfigures progress in magnitude of the exploration challenge (height of the Boulder). Left: Both intrinsic motivation methods quickly solve the low Boulder of height 10, with $\epsilon$-greedy also catching up eventually. Middle: For the middle height Boulder of size 30, reward-based intrinsic motivation starts to suffer, while $\epsilon$-greedy cannot solve it at all within 20k steps. Right: For the high Boulder of size 100, only goal-based intrinsic motivation manages to solve it. All agents use Q-learning, results averaged over 10 repetitions.

approaches on Boulders of varying heights. The left, middle, and right graph display results for Boulders of height 10, 30 and 100 respectively, which correspond to a light, medium and heavy sparse reward. On the left, we see that an *undirected* exploration method based on random perturbation (Mozer & Bachrach, 1989), such as $\epsilon$-greedy action selection (blue solid line), does solve the low Boulder (left graph), but fails on the medium (middle) and high (right) one. An intrinsic motivation method based on a novelty reward bonus (yellow dashed line) (Chentanez et al., 2004) partially overcomes this problem, and manages to also solve the medium Boulder. However, it also suffers on the high Boulder, mostly because of the 'detachment' problem (the agent has discovered the goal, but the novelty reward has not been fully propagated yet) (Ecoffet et al., 2021). Finally, in the right graph with heavy exploration challenge, we see that only goal-based intrinsic motivation methods (Colas et al., 2021), such as Go-Explore (Ecoffet et al., 2021), manage to solve the task because it explicitly learns to return to interesting states. The full notebook further illustrates the difference between these methods, for example showing visitation density maps over the full Boulder.

**Partial observability in the MemoryCorridor** In the interactive notebook on partial observability, which uses the MemoryCorridor environment, we for example generate the graph on the left of Figure 3. Here, we study how a tabular agent can use *framestacking* (or *windowing* (Lin & Mitchell, 1992; McCallum, 1997), i.e., concatenating the most recent observations into a joint state) to further advance through the corridor. The framestack allows the agent to remember which door we need to select at every step. The default agent without any memory (framestack = 1, blue dashed line) does not get very far, because it has no memory and therefore essentially has to guess at every step. When we do include more historical observations (yellow, green and red lines), the agent's performance clearly starts to improve, reaching just above the length of its memory (the bonus is because of the chance to guess correctly afterwards). We do however see that learning starts to progress more slowly, because the size of the required Q-table grows, and we need more samples to fill it with appropriate estimates. The framestacking approach appears successful, but it has also its limitations. Due to the curse of dimensionality, the size of the table that we need to store increases very quickly with the length of the window. In the notebook, we show how a framestack of length 25 already requires over a petabyte ($10^{15}$) of data, making the approach no longer feasible. We then show how function approximation, for example, based on deep learning, may overcome this limitation and scale to longer historical dependency (due to the effect of generalisation).

**Model-based reinforcement learning in the Supermarket** The interactive notebook on model-based reinforcement learning, which uses the Supermarket environment, includes the right graph of Figure 3. The figure compares model-free Q-learning (blue solid line) (Watkins & Dayan, 1992) to two classic model-based RL methods: Dyna (yellow dashed line) (Sutton, 1990) and Prioritised Sweeping (green dotted line) (Moore & Atkeson, 1993). We see both model-based RL methods learn faster than model-free RL due to the planning updates of the solution. Interestingly, Prioritised Sweeping eventually reaches a lower final performance. It turns out it propagates information so fast that it 'misses' the third item in the supermarket (a better exploration method would of course

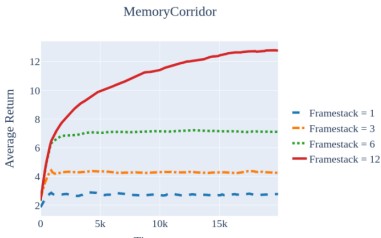 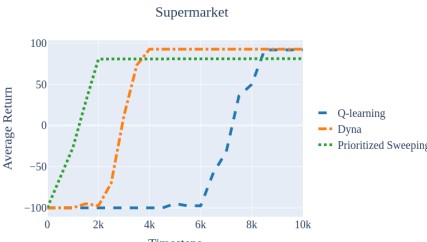

Figure 3: Left: Illustration of *framestacking/windowing* to overcome partial observability on the MemoryCorridor. Agents that include more historical information in their state manage to advance further in the corridor. All agents use Q-learning, results averaged over 10 repetitions. Right: Illustration of two model-based RL algorithms (Dyna and Prioritised Sweeping) versus model-free RL method (Q-learning) on the Supermarket. Both model-based RL methods learn faster since they use a learned model to more effectively propagate information through the value function solution. All agents use $\epsilon$-greedy exploration, results averaged over 10 repetitions.

solve this issue.) The notebook on model-based reinforcement learning also illustrates the effect of different planning budgets on the performance of these model-based algorithms, explains and implements other planning methods such as Dynamic Programming (Bellman, 1954), and displays the influence of model noise/uncertainty on planning performance.

## 6   STUDENT EVALUATION

We empirically evaluated EduGym among bachelor/master students who had previously taken a reinforcement learning class and RL researchers (mostly early PhD students) who attended a reinforcement learning workshop. This way, participants were not completely novel to the field and could judge whether EduGym added to their current understanding of reinforcement learning. We found 89% (31 out of 35) of participants think EduGym improves their conceptual understanding of reinforcement learning, while 77% (27 out of 35) judge EduGym improves their practical understanding. Overall, 86% (30 out of 35) of participants consider EduGym to be a useful tool for reinforcement learning education. Detailed results of the questionnaire are available in Appendix B.

## 7   DISCUSSION

This paper introduced *EduGym*, an educational suite for reinforcement learning. First of all, EduGym consists of a set of environments that 1) are designed for specific challenges in reinforcement learning, and 2) have customizable parameters to vary the specific challenge. In addition, it contains a set of interactive notebooks that 1) gradually introduce the specific challenge and possible solutions, integrating explanations, equations and code, and 2) allow for active experimentation by the student. Evaluation among students indicates EduGym is considered a useful tool for reinforcement learning education, and we hope it will as such benefit both students and educators in the field.

Reinforcement learning research mostly focuses on 'deep' implementations, while EduGym contains several low-dimensional environments and tabular algorithms. However, this is on purpose, since it allows for faster experimentation and better interpretability, which is what we require for educational purposes. Deep learning is itself specifically covered in the notebook on state dimensionality, while several other notebooks (e.g. on partial observability, continuous state/action spaces) also contain deep implementations in their later stages.

There are several reinforcement learning challenges we have not included in EduGym, such as multi-agent reinforcement learning (Zhang et al., 2021), multi-objective reinforcement learning (Hayes et al., 2022), goal-based reinforcement learning (Schaul et al., 2015), etc. We intentionally chose to first focus EduGym on the key concepts of 'standard' single-agent reinforcement learning, to let students get familiar with this setting. However, we will keep evaluating EduGym in courses and continuously improve and expand the library based on feedback.

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

## A EDUGYM ENVIRONMENTS

This section details each of the environments included in EduGym. Per environment, we cover the general idea of the environment, the parameters that can be varied, and detail the MDP formulation (state space, action space, dynamics, reward function and termination criteria).

### A.1 BOULDER

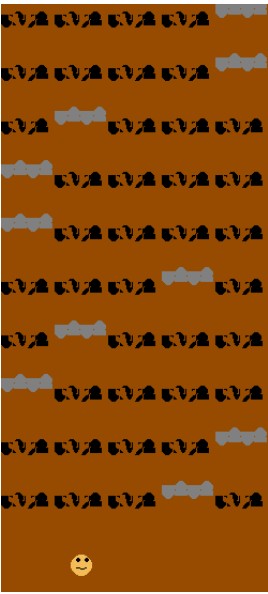

- **Designed for**: Exploration (under sparse reward).
- **Description**: An agent (visualised as a yellow smiley face) is bouldering and aiming to reach the top of the wall. The agent needs to take a sequence of right actions without making any wrong decisions in order to reach the top grip. A wrong action will take the agent back to the initial state. Correct grip points are visualised as grey solid stones while black cracked stones indicate wrong actions.
- **Variable parameters**:
    - *Height* ($H$): the length of the sequence of actions the agent has to perform correctly to reach the top.
    - *Number of grips* ($N$): larger $N$ makes the size of the action space larger.

    Note that the average number of steps needed to reach the top under random action selection is $N^H$. Increasing the height $H$ makes the challenge therefore *exponentially* harder for undirected exploration.
- **State space**: `Discrete(H)`. The current position (height) of the agent on the wall.
- **Action space**: `Discrete(N)`. The index of the grip the agent will jump to next.
- **Dynamics**: If the taken action matches the grip the agent will move upward, otherwise the agent falls back to the initial state.
- **Reward function**: A 1 when the agent successfully reaches the top, otherwise the reward will always be 0.
- **Initial State**: The bottom of the wall.
- **Termination**: When the agent successfully reaches the top or the maximum number of steps is reached.
- **Compared algorithms (notebook)**: Q-learning with $\epsilon$-greedy (Sutton & Barto, 2018), Q-learning with count-based intrinsic reward (Chentanez et al., 2004), Go-Explore (Ecoffet et al., 2021).

## A.2   ROADRUNNER

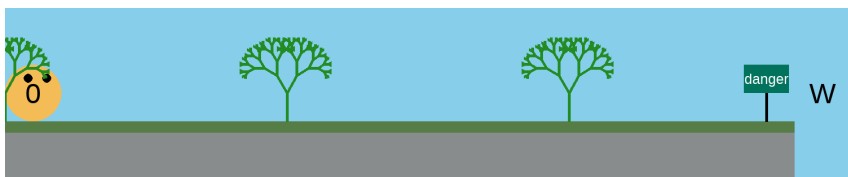

- **Designed for**: Credit assignment: on- versus off-policy.

- **Description**: In the Roadrunner environment, an agent is tasked to run towards a cliff as quickly and closely as possible, *without actually falling in*. The agent moves along a 1-D plane, starting from position 0 up to a variable target position $T = W - 1$. The goal of the agent is to arrive exactly at the target position. Moving beyond the target position means that the agent falls off a cliff, thus losing the game. The agent only has the option to increase or decrease its current speed, starting from zero. The agent needs to learn how to control its speed appropriately. Similar to missing the target position, slowing down too much also causes the agent to lose the game.

- **Variable parameters**:
    - *Environment size* (`W`): How many cells the environment consists of.
    - *Negative reward size* (`R`): How much reward is given when the agent 1) falls off the cliff or 2) slows down beyond a speed of 0.
    - *Max speed* (`MAX_SPEED`): what the maximum speed is of the agent, which we need to limit to make tabular learning feasible.

- **State space**: `MultiDiscrete(W,MAX_SPEED)`. The two state elements denote $\{x, dx\}$, with $x$ the current location of the agent, and $dx$ the current speed of the agent.

- **Action space**: `Discrete(3)`. The three actions $\{-1, 0, +1\}$ denote the applied change to the agent's speed ($dx$).

- **Dynamics**: Every step the agent's location ($x$) is updated with its speed ($dx$). If the agent moves beyond `W`, the agent is reset to `W`.

- **Reward function**: -1 at every step. Reaching the target position `T` gives +1. Hitting `W` gives `R` (default is -100). Slowing down below a speed of 0 also gives `R`.

- **Initial State**: $x = 0, dx = 0$

- **Termination**: Agent reaches `T`, or its speed drops below 0.

- **Compared algorithms (notebook)**: Tabular Q-learning (Watkins & Dayan, 1992) and SARSA (Rummery & Niranjan, 1994).

## A.3 STUDY

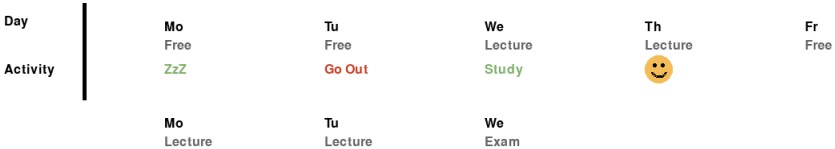

- **Designed for**: Credit assignment: depth.
- **Description**: This environment aims at simulating the study efforts of a student for an exam. Each day, the student can decide to either: *Study* for the exam, *Sleep* to regain energy, *Go Out* at the cost of their energy level, or do any *Other* action with no particular effect towards the final exam. For this specific course, studying for the exam is only effective on a day with a lecture. The total number of days before the exam can be set as a hyperparameter. In order to pass the exam, the student must study on enough lecture days and have enough energy to actually take the exam.
- **Variable parameters**:
  - *Number of actions:* This directly controls the variance of trajectories. Smaller values make $n$-step work better with larger $n$, and vice versa.
  - *Action reward noise mean:* This sets the range of the mean rewards for all actions. At initialisation, a reward between the negative of this value and 0 is randomly drawn for each action.
  - *Action reward noise sigma:* This parameter introduces variance on all action rewards and can be used to change the variance of the returns.
  - *Total days:* The total number of days until the exam. This is effectively the episode length and also a means of adapting the variance.
  - *Lecture days:* The number of lectures that will take place before the exam.
  - *Lectures needed:* The minimum number of lectures that are needed to have enough knowledge for the exam.
  - *Energy needed:* The minimum energy level that is needed to pass the exam.
- **State space**: `MultiDiscrete(5,5,H)`. At any time $t$, the agent has a certain knowledge $k$ and energy level $e$. We, therefore, have a 3D state space with a state $s = (k, e, t) \in \mathcal{K} \times \mathcal{E} \times \mathcal{T}$, where $\mathcal{K} = \mathcal{E} = \{0, ..., 5\}, \mathcal{T} = \{0, ..., H\}$ and $H$ is total number of days.
- **Action space**: `Discrete(N)`. $A = \{Study, Sleep, GoOut, Other_1, Other_2, ..., Other\_N\}$, where $N$ is the number of other actions the agent can take.
- **Dynamics**: Studying on a lecture day increases the knowledge level $k$. Sleeping increases the energy level $e$. (until a maximum of 4). Going out decreases the energy level (until a minimum of 0). Any other action has no effect.
- **Reward function**: The reward is 10, if the student studies on the exam day and $k \geq K$ and $e > E$, where $K$ and $E$ are the number of lectures needed and the amount of energy needed, respectively. Furthermore, every action has a reward that is sampled from $N(r_i, \sigma)$ where $r_i$ is the reward noise mean of action $i$ and $\sigma$ is the reward noise sigma.
- **Initial State**: At time 0, the agent has no knowledge and no energy, hence $s_0 = (0, 0, 0)$.
- **Termination**: An episode ends when the exam day is reached. This is at time step $H$.
- **Compared algorithms (notebook)**: $n$-step SARSA (from 1-step to Monte Carlo) Sutton & Barto (2018).

## A.4 CATCH

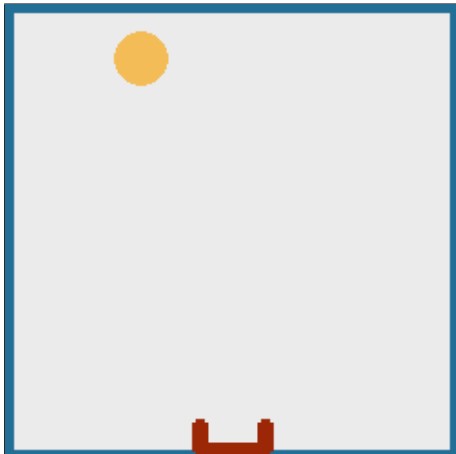

- **Designed for**: State dimensionality.
- **Description**: The agent controls a paddle at the bottom of the screen and needs to catch a ball dropping from the top.
- **Variable parameters**:
    - *Grid size* (`rows` & `columns`): Influences range of values within an observation type and its memory size.
    - *Observation Type* (`observation_type`): Determines the observation space. Choice between: `vectorised`, `default` (grid), `rgb`.
- **State space**:
    - Vectorised: `MultiDiscrete(rows,rows,columns)`: The $x$ and $y$ of the ball, and the $x$ of the paddle.
    - Grid: `Box(0,1,[rows,columns])`. A grid filled with zeros with a 1 at the paddle and a 1 at the ball.
    - Image: `Box(0, 255, [rows,columns,3])`. A grid with RGB values for each location.
- **Action space**: `Discrete(3)`. Agent can move left, right or stay put.
- **Dynamics**: The ball drops in increments of one along the y-axis.
- **Reward function**: A 0 at every step, except for a +1 at the end of the episode +1 if the ball was caught or -1 if it was not.
- **Initial State**: The player/paddle at the bottom always starts in the middle. The ball drops randomly somewhere from the top.
- **Termination**: The ball reaches the bottom (the height of the player/paddle).
- **Compared algorithms (notebook)**: Tabular Q-Learning  Watkins (1989), Deep Q Learning Mnih et al. (2013) (Stable Baselines 3  Raffin et al. (2019))

## A.5 MEMORYCORRIDOR

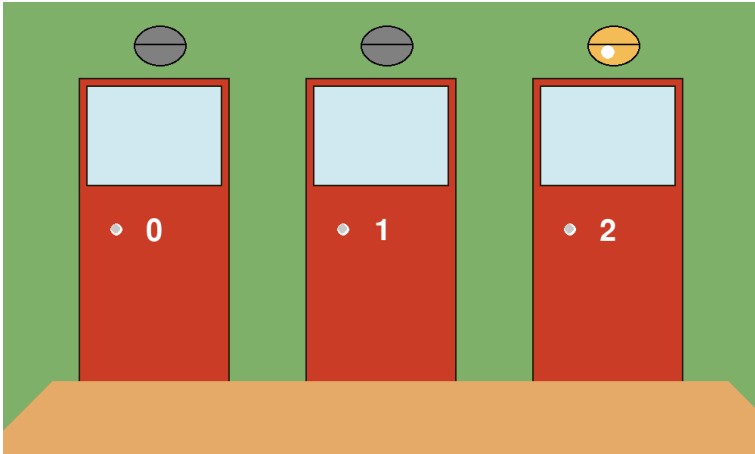

- **Designed for**: Partial observability (memory).

- **Description**: The agent has to navigate a sequence (corridor) of doors. The agent is shown a set of doors in each step of the sequence. Only one of the observed doors advances the agent to the next step in the sequence. Opening any other door terminates the episode. Each time the sequence is completed, the sequence repeats with an increased length of one. Only on the final set of doors in the sequence, the agent is shown what the correct door to go through is. In all other states, the agent observes no useful information and has to open the correct door based on information obtained in prior states. The task automatically scales in its memory challenge due to the length of the sequence increasing when the agent does well.

- **Variable parameters**:

  - *Number of doors* (`num_doors`): The number of doors in each state of the sequence. This increases the state space, but not the difficulty of memorization for the agent. This difficulty is scaled by the environment each time a sequence is completed successfully.

- **State space**: `Discrete(num_doors + 1)`. A state for each of the doors being the correct door or no correct door visible.

- **Action space**: `Discrete(num_doors)`. Each action corresponds to opening a door (open door 0, open door 1 etc.)

- **Dynamics**: Every action opens one of the available doors. If this is the correct door, the agent proceeds. If the door is the final door of the sequence, the agent has to replay the entire sequence with one additional door at the end.

- **Reward function**: Opening the correct door rewards the agent with 1, else no reward is given.

- **Initial State**: The correct door is observed in the first state as it is the final door of the first sequence, which is of length 1.

- **Termination**: Termination occurs when the agent opens an incorrect door.

- **Compared algorithms (notebook)**: Tabular Q-learning with framestacking, Deep Q Learning with framestacking (Stable Baselines 3 Raffin et al. (2019)) and Recurrent PPO (Stable Baselines 3 - contrib Schulman et al. (2017))

### A.6    TAMAGOTCHI

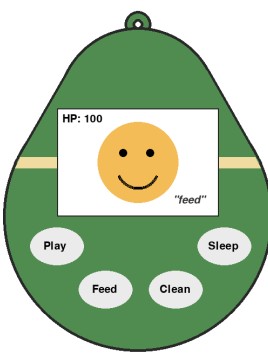

- **Designed for**: Amount of state signal (& language).
- **Description**: The agent needs to take care of a Tamagotchi, for which it has four actions available: 'play', 'sleep', 'feed', or 'clean'. The Tamagotchi has four internal variables, each between 0 and 100, that indicate how much the agent currently needs each of these actions. Each action increases its associated variable by +30, while decreasing the other variables by -5. However, the agent cannot directly observe these internal variables. Instead, it only sees the Tamagotchi's HP, which is derived from the combination of internal variables. However, the Tamagotchi also communicates a message about its internal state (essentially indicating which variable is deficient). By modifying the amount of noise in this message at environment initialization, we can control the amount of information the observation provides about the true internal Tamagotchi state.
- **Variable parameters**:
  - Message noise `tau`: Temperature parameter that influences the amount of noise in the message. For `tau` $\to 0$ the message becomes a perfect description of the internal variables, for `tau` $\to \infty$ it becomes pure noise.
  - Message size `max_msg_length`: length of the utterances.
  - Vocabulary size `H`.
- **State space**: `MultiDiscrete(100,H,..)`, where the first variable indicates the HP level, and the second (or more) variables indicate the communicated message, out of a vocabulary of size `H`.
- **Action space**: `Discrete(4)`. Play, sleep, feed and clean.
- **Dynamics**: The actions of the agent influence the internal variables of the Tamagotchi. Each action results in a positive update (+30) of the corresponding internal variable and a small decrease (-5) of the other variables. When the action is not the ideal action according to the Tamagotchi, all variables (thus including the one that has just been updated) will decrease by -10. After the variables have been updated the Tamagotchi decides which action should be taken next (through weighing the importance of each internal variable) and generates an utterance of a specified length accordingly. As described above, these utterances can be very informative or noisy.
- **Reward function**: Ranges in $[-200, 75]$. The reward corresponds to the happiness of the Tamagotchi. This is calculated based on its internal variables, where lower values get a higher relative weight (contribute more to the aggregated reward).
- **Initial State**: $\{100, U_1, ..., U_n\}$ where $U$ denote tokens in the utterance of the Tamagotchi, and `n = max_msg_length`.
- **Termination**: `HP = 0` or the maximum steps per episode (`steps_per_episode == 100`) is reached.
- **Compared algorithms (notebook)**: Q-learning with variation in noise.

## A.7 TRASHBOT

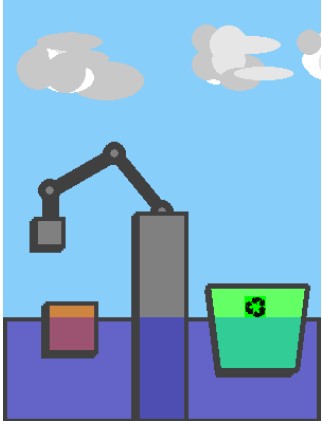

- **Designed for**: State/action spaces, discrete versus continuous.
- **Description**: The agent (a robotic arm with 2 degrees of freedom) has to move red boxes to the container (green box). The main idea illustrated by this environment is that the discretisation of continuous action spaces into an appropriate number of unique actions increases sample efficiency. Additionally, a lower number of possible actions further speeds up learning, as long as the precision level that is necessary to achieve the task is not compromised.
- **Variable parameters**:
    - **Discretization level** - when discrete action space is used, determines how many bins the continuous action space is segregated into.
    - **Width of the container** - a smaller green container makes the problem harder by increasing the level of precision required in order to obtain the final reward.
- **State space**: `Continuous (7)` - **(4)** current positions of motors joints, **(2)** coordinates of the magnet, **(1)** binary variable indicating whether the arm is holding a box.
- **Action space**:
    - `Continuous(2)` - affects the change in joint angles in radians, range $[-1 : 1]$.
    - `Discrete(`*num_bins*`)` - the same continuous range discretised into $[3/5/7/9/11]$ bins.
- **Dynamics**: At every time step the two controllable angles of the motors are updated based on the selected action values. Collision checks between the agent and the rest of the environment are performed to grant rewards and to check for terminal conditions.
- **Reward function**: +1 reward for picking up a red box. +2 reward for putting the red box into the green box. Up to +2 for placing the red box at the exact centre of the green container (incentive for precise movement). -2 for colliding with the other environment objects. -1 for exceeding the max step limit (150).
- **Initial State**: The initial state of the environment is shown in the figure above.
- **Termination**: Whenever the agent or the red box collides with other objects; maximum step count is reached (150).
- **Compared algorithms (notebook)**: Proximal Policy Optimisation *(PPO)*Schulman et al. (2017) - continuous and discrete formulations.

## A.8 GOLF

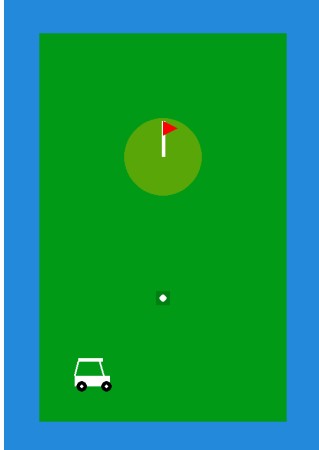

- **Designed for**: Dynamics, stochasticity.

- **Description**: The ball (white dot) starts on the bottom side of the golf course. At each step, the agent hits the ball with a chosen type of swing. The goal of the agent is to get the ball on the green after which the game is finished and the agent gets a reward. Reaching the maximum number of allowed hits or hitting the ball off course also causes the game to end, but the agent gets a negative reward instead. A stochasticity level is used to mimic the skill level of the agent. It causes the ball to deflect from the coordinates the ball was hit towards.

- **Variable parameters**:
  - *Stochasticity level* (`stochasticity_level`): Used to set the standard deviation of the zero-centred Gaussian noise that is used to sample the deflection of the ball ($\sigma = $ `stochasticity_level` $*$ `action`$^2$).

- **State space**: `MultiDiscrete(width, length)`. The coordinates of the ball on the field.

- **Action space**: `Discrete(3)`. The action can be a drive, chip, putt or i.e. a long shot, a medium shot, and a short shot.

- **Dynamics**: The ball moves forward towards the flag with a distance depending on the type of swing performed. A random deflection in the (transverse) direction of the movement where the magnitude of the deflection is defined through the stochasticity level.

- **Reward function**: -1 is obtained if the ball goes off the course or if the maximum number of hits is reached. If the green is reached, the reward is between 0 and 1 proportionally to the number of needed hits.

- **Initial State**: The ball starts at (width / 2, 0).

- **Termination**: When the ball is off course, when the ball is on the green, and when the maximum number of hits is reached.

- **Compared algorithms (notebook)**: Q-learning, risk-sensitive Q-Learning (Delétang et al., 2021).

## A.9  SUPERMARKET

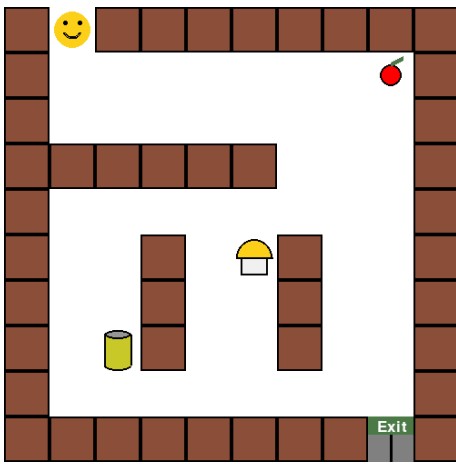

- **Designed for**: Model-based reinforcement learning.
- **Description**: The agent (yellow square) enters a supermarket, where at each step it can move up, down, left or right. It has three items on its shopping list, indicated by the red, blue and green squares. The agent can always exit the supermarket at the bottom right. To mimic the cost of actions in the real world, each `step(a)` takes some time to finish (see variable parameters). However, the agent also has access to `model(s,a)` of the environment, which executes without an extra time penalty.
- **Variable parameters**:
  - *Step time-out* (`step_timeout`): interval the environment blocks between subsequent calls to the `step()` method.
  - *Model noise* (`noise`): standard deviation of zero-centred Gaussian noise added to the reward model.
- **State space**: `Discrete(800)`. The unique state is identified based on agent (x,y) location plus the collection status of the three items.
- **Action space**: `Discrete(4)`. The agent can move up, down, left and right.
- **Dynamics**: The agent moves in the specified direction unless it hits a wall. It automatically collects items when it steps on them.
- **Reward function**: A -1 default penalty on each step, with added +25 for each collected item, and added +50 for leaving the supermarket (bottom-right).
- **Initial State**: Top-left, visible in the figure at the yellow circle.
- **Termination**: Bottom-right, visible in the figure as the bottom-right door.
- **Compared algorithms (notebook)**: Dynamic Programming, Dyna, Prioritised Sweeping.

# B   STUDENT EVALUATION

We empirically evaluated EduGym among bachelor and master students who had previously taken a reinforcement learning class, and among researchers who attended a reinforcement learning graduate course. These candidates received an email with the invitation to try out EduGym (through a link to the public website) and afterwards fill in an anonymous questionnaire. The questionnaire consisted of three statements about the conceptual, practical and overall benefits of EduGym (exact statements shown in Table 2). Each of these statements had to be evaluated on a 5-point Likert scale. In addition, participants could provide open-text feedback, about parts of EduGym they liked and/or suggestions for improvement.

In total 35 participants replied to the questionnaire, of which 46% was currently a PhD student, 37% was a bachelor/master student, and 17% did not fit these categories (either a more senior RL researcher or a student that has graduated from university). The questionnaire results are shown in Table 2. We observe 89% (31 out of 35) of respondents think EduGym improves their conceptual understanding and 77% (26 out of 35) think EduGym improves their practical understanding. Overall, 86% (30 out of 35) (strongly) agree that EduGym is a useful tool for reinforcement learning education. One particular respondent did not like EduGym at all and replied 'strongly disagree' to all statements. The questionnaire did contain the option of free textual feedback, but this student did not leave any textual response. Since the questionnaire was fully anonymous (to not influence the participant's judgement), we could not find out why this particular respondent did not like EduGym.

Table 2: Results of the EduGym evaluation questionnaire. Each cell displays the total number of respondents that ticked the specific box. In total 35 participants completed the questionnaire.

|  | Strongly disagree | Disagree | Neither agree nor disagree | Agree | Strongly agree |
|---|---|---|---|---|---|
| EduGym improves my *conceptual* understanding of reinforcement learning | 1 | 1 | 2 | 27 | 4 |
| EduGym improves my *practical* understanding of reinforcement learning | 2 | 4 | 2 | 20 | 7 |
| I think EduGym is a useful tool for reinforcement learning education | 1 | 0 | 4 | 13 | 17 |

