# OpenReview forum: "EduGym: An Environment Suite for Reinforcement Learning Education"
_ICLR.cc/2024/Conference — Submitted to ICLR 2024_

### Official Review · Reviewer_FCg2 · 2023-10-19

**Soundness:** 3 good
**Presentation:** 4 excellent
**Contribution:** 3 good
**Rating:** 6
**Confidence:** 3

**Summary:**

This paper proposes a novel suite, EduGym, for guiding students to learn about RL. EduGym proposes several simple environments that demonstrate the unique challenges in RL. It also provides notebooks for a quick start on these environments. The authors interviewed junior graduate students about EduGym, and most students think that it help them improve understanding in RL.

**Strengths:**

Learning RL is always a hard task for junior graduate students, as it requires a lot of basic knowledge (MDPs, dynamics programming, etc.), as well as broad knowledge about current progress and challenges in DRL.  Although there have been textbooks like RL: An Introduction, it mainly focuses on traditional RL, and has few discussions about DRL. Meanwhile, there have been some people who make RL lectures online, but they can hardly contain all the critical challenges in DRL. EduGym serves as a good first step towards a powerful toolkit that enables a direct demonstration to current DRL challenges.

**Weaknesses:**

The authors do not provide information about whether open-sourcing their codes. EduGym has the potential of becoming a powerful toolkit for starting RL research, if it can be jointly maintained by the RL community.

**Questions:**

1. Do the authors plan to open-source the codes, and invite the community to jointly maintain the codebase?
2. Do the authors plan to add more examples on more RL methods like target network, double Q networks, etc?

---

> ### Author Response · Authors · 2023-11-17
> **Open-source and contributing**
>
> To answer your questions:
>
> * Yes we definitely open-source the code, and invite everyone to contribute. The repository is already on Github, although we kept it private for now because of the double-blind review.
>
> * We will add novel methods in the future, and have been discussing an “EduGym 2”. Topics to include could be multi-agent, multi-objective, goal-conditioned, etc., as we discuss in the last paragraph of the Discussion. Your suggestions are also very valid!
>
> We hope this answers your questions. We appreciate your summary under strenghts, which indeed captures our exact intentions. We are curious to hear whether the additional feedback affects your judgement.

---

> > ### Comment · Reviewer_FCg2 · 2023-11-20
> > **Thank you for your Response**
> >
> > I thank the authors for addressing my concerns. Although I do not have further questions about the contents of this paper and appreciate its presentation and contributions, I am aware of other reviewers' concerns about whether EduGym is suitable for the main track of the conference. I think it's okay, but I will also consider AC's opinion. I maintain my current score.

---

### Official Review · Reviewer_16ha · 2023-10-27

**Soundness:** 2 fair
**Presentation:** 2 fair
**Contribution:** 1 poor
**Rating:** 3
**Confidence:** 5

**Summary:**

This paper introduces a library named EduGym, which is a set of educational RL environments and interactive notebooks. It aims address the challenges students  encounter in studying RL. With EduGym, students can translate between equations and code in a cohesive manner by interactive notebooks that explain the challenges and their potential solutions. An evaluation conducted among RL students and researchers indicates that 86% of them find EduGym to be a valuable tool for reinforcement learning education.

**Strengths:**

**Originality:**

EduGym is an educational library for RL students. Its key novelty are: (1) it provides a set of educational RL environments each specifically designed for a particular RL challenge; (2) to better illustrates RL concepts, it a set of interactive notebooks, where students are gradually taught about each challenge, and can at the same time actively experiment with relevant code.

**Quality:**

This paper developed many RL environments (e.g., exploration, partial observability, stochasticity, etc.) and provided high-quality code for students to learn RL.

**Clarity:**

This paper has a clear motivation and the whole paper is easy to follow.

**Significance:**

In the educational context, this paper has a positive impact on teaching RL.

**Weaknesses:**

Despite the merit of this paper, weaknesses of this paper are:

1. There is no key contribution to RL algorithm and research. This paper introduces an educational library for RL learners. Its key contribution is limited to RL teaching, which is not the key scope of ICLR conference.
2. There are many excellent libraries for RL researchers and learners, such as OpenAI’s Spinning Up library and the code of Sutton and Barto’s RL book. Spinning UP provides simple and highly-consistent code for (Deep) RL researchers and practitioners. The code of Sutton and Barto’s RL book also provides example for students to understand concepts, such as the Markov chain example in Q-learning and Double Q-learning. Environments in EduGym are deliberately designed but are not conceptually novel in RL research. EduGym does not provide any guidance for RL practitioners.
3. It lacks best practices on how to use deep neural networks in RL.

**Questions:**

What sets EduGym apart from Sutton and Barto’s RL book and its code?

---

> ### Author Response · Authors · 2023-11-17
> **Discussion of comparison**
>
> You indicate our work is outside the scope of ICRL, which we disagree with. In recent years, many papers have appeared at main machine learning conferences that present either an environment suite, benchmark and/or educational effort. Examples include [1,2,3,4,5]. Our work actually falls in the same category. We agree these papers don’t present a novel algorithm, but we disagree they don’t contribute to the research community (they usually get cited a lot). The research community also needs educational efforts for new students entering the field, as does it need benchmarks and small test environments that isolate a particular challenge (which did not exist yet as a collection). Since this type of work has appeared at ICLR, ICML and NeurIPS over recent years [1,2,3,4,5], we do not think it is fair to judge our paper as outside the scope of ICLR.
>
> You also ask about the difference with OpenAI SpinningUp and the code of Sutton and Barto. Spinning Up is very different: it is educational (and very useful), but it implements complex specific deep RL algorithms, such as PPO and DDPG. This is very useful for more advanced students, but it does not isolate individual conceptual aspects of RL, such as partial observability, on/off-policy, sparse exploration, etc. We write about this in the Related Work section, in the paragraph called “Algorithmic frameworks”.
>
> The code of Sutton and Bart comes much closer, and the CliffWalking example to illustrate the difference between on/off-policy was a major inspiration. However, there are two main differences: 1) Sutton and Barto mostly focus on credit assignment, and do not include specific conceptual examples for most of the challenges we did include (state dimensionality, partial observability, language, continuous actions) etc., and 2) most importantly, the code of Sutton and Barto does not come with interactive notebooks, which prevents the mixing of text and code, and makes it way less accessible to students.
>
> Finally, you aks about best practices on deep learning. We do include a notebook about deep learning, which is the notebook on state dimensionality. Also, deep learning appears in several other notebooks, such as partial observability and continuous state/actions. However, we do indeed not discuss all practical aspects of tuning deep learning, but this is also clearly beyond the scope of our work. We intend to teach reinforcement learning fundamentals: exploration, credit assignment, partial observability, etc. Our main focus lies on small conceptual examples, while for complex deep RL many codebases actually do exist (such as Spinning Up).
>
> We are curious to hear your view on the above topics.
>
>
> 1. Osband, I., Doron, Y., Hessel, M., Aslanides, J., Sezener, E., Saraiva, A., ... & Van Hasselt, H. (2019, September). Behaviour Suite for Reinforcement Learning. In International Conference on Learning Representations.
> 2. Liang, E., Liaw, R., Nishihara, R., Moritz, P., Fox, R., Goldberg, K., ... & Stoica, I. (2018, July). RLlib: Abstractions for distributed reinforcement learning. In International conference on machine learning (pp. 3053-3062). PMLR.
> 3. Pleines, M., Pallasch, M., Zimmer, F., & Preuss, M. (2022, September). Memory Gym: Partially Observable Challenges to Memory-Based Agents. In The Eleventh International Conference on Learning Representations.
> 4. Terry, J., Black, B., Grammel, N., Jayakumar, M., Hari, A., Sullivan, R., ... & Ravi, P. (2021). Pettingzoo: Gym for multi-agent reinforcement learning. Advances in Neural Information Processing Systems, 34, 15032-15043.
> 5. Yu, T., Quillen, D., He, Z., Julian, R., Hausman, K., Finn, C., & Levine, S. (2020, May). Meta-world: A benchmark and evaluation for multi-task and meta reinforcement learning. In Conference on robot learning (pp. 1094-1100). PMLR.

---

### Official Review · Reviewer_iCzW · 2023-10-29

**Soundness:** 3 good
**Presentation:** 3 good
**Contribution:** 2 fair
**Rating:** 6
**Confidence:** 4

**Summary:**

This paper introduces EduGym, an educational tool for reinforcement learning, designed to address the challenges students face in understanding and applying RL concepts. It provides a set of educational RL environments, each tailored to isolate specific RL challenges, along with interactive notebooks that bridge the gap between equations and practical code, aiding in faster learning progress.

**Strengths:**

- The paper addresses a real-world issue faced by students entering the field of reinforcement learning, making it highly relevant and practical for both educators and learners.
- EduGym introduces a novel solution by providing a set of educational RL environments and interactive notebooks, which can enhance the understanding of RL concepts through hands-on experimentation.
- EduGym provides comprehensive coverage over various aspects of RL.

**Weaknesses:**

- The background section seems to be irrelevant as it hasn't been used, referred to, or elaborated in the rest of the paper.
- The lack of introduction to basic RL algorithms like value iteration or REINFORCE [1].
- Lack of innovation.

[1] Williams, Ronald J. "Simple statistical gradient-following algorithms for connectionist reinforcement learning." Machine learning 8 (1992): 229-256.

**Questions:**

- It would be beneficial to include information about the pedagogical principles or educational theories that guided the development of EduGym. How was the content structured to support effective learning, and were any instructional design models considered?
- While EduGym is presented as a solution to the challenges in RL education, are there any plans to expand its content, incorporate new challenges, or update the environments and notebooks in the future to keep pace with evolving RL research?
- What is the plan for maintaining and updating EduGym to keep it relevant and aligned with the latest developments in RL research and education?

---

> ### Author Response · Authors · 2023-11-17
> **Discussion on raised questions**
>
> You first of all ask about the learning theory below EduGym. This is mostly “Learning by Doing”, about which there is much literature [1,2,3]. The main idea is that we need to connect theory (equations, text) to practical interactive examples (code, notebooks) in a single document. In addition, we attempt to break the problem down in underlying conceptual topics. We have not included this in the paper since it seemed too far off for a machine learning conference, but of course see your point as well, and could include it.
>
> You also ask whether EduGym will be 1) extended and 2) kept up-to-date. Regarding the former, we definitely will extend Edugym, as we indicate in the last paragraph of the Discussion, where we for example indicate we think of a future EduGym extension with multi-agent, multi-objective, goal-conditioned RL, etc. Regarding the latter, we will also maintain EduGym, but additions will probably mostly focus on novel notebooks. The reason is that EduGym teaches the RL basics, such as the distinction between on- and off-policy updates. These conceptual topics are so fundamental to RL, that their basic explanation will probably stay quite stable. Of course, with novel research completely novel topics may develop, which we will then try to incorporate into EduGym.
>
> As a weakness you mention the lack of value iteration and policy gradients. They are actually included: value iteration appears in the notebook on model-based RL, while the policy gradient appears in the notebook on continuous state/actions.
>
> You finally mention that the innovation is limited, which we disagree with. It’s true that EduGym does not contribute a novel algorithm, but we believe the RL community as a whole has immensely profited in recent years from environment suites, benchmarks and educational efforts. These papers often get quite some traction (citation-wise), which indicates they have their merit for the community. We think EduGym also falls in this category of papers, and although there is no novel algorithm, we do not think this implies the innovation is limited.
>
> You otherwise seem quite positive about the idea of EduGym, and are curious to hear your thoughts on our comments.
>
>
> 1. Schank, R. C., Berman, T. R., & Macpherson, K. A. (2013). Learning by doing. In Instructional-design theories and models (pp. 161-181). Routledge.
> 2. Thompson, P. (2010). Learning by doing. Handbook of the Economics of Innovation, 1, 429-476.
> 3. Anzai, Y., & Simon, H. A. (1979). The theory of learning by doing. Psychological review, 86(2), 124.

---

> > ### Comment · Reviewer_iCzW · 2023-11-22
> > **Response to authors**
> >
> > I would like to thank the authors for their clarifications. My concerns and questions were mostly addressed. After going through the discussions between the authors and other reviewers, I intend to increase my rating from 5 to 6.

---

### Official Review · Reviewer_pfPX · 2023-11-01

**Soundness:** 3 good
**Presentation:** 2 fair
**Contribution:** 3 good
**Rating:** 3
**Confidence:** 3

**Summary:**

The paper introduces EduGym, a suite of educational reinforcement learning environments and interactive notebooks designed to address the struggles faced by students in understanding and applying reinforcement learning concepts. EduGym provides specific environments that illustrate different challenges in reinforcement learning, such as exploration, partial observability, and stochasticity, allowing students to gain practical experience. It simplifies the learning process, provides insight, and offers a quick experimentation loop for students, catering to their educational needs.

**Strengths:**

- I can see the big effort of the authors to build EduGym and kind motivation to help students to learn reinforcement learning.
- Providing interactive notebooks that explain the challenges, possible solution approaches, and experimental performance of each environment.

**Weaknesses:**

However, my main concern is that this paper is not a research paper w.r.t. solving existing research problems or proposing new problems. This paper is more like a piece of description or instruction. This paper should be submitted to the dataset/benchmark track or blog post track, instead of the main track of ICLR.

**Questions:**

I went through the provided Jupyter Notebook and found the authors did a really good job for the RL beginner. However, I still suggest the authors submit this paper to other tracks.

**Details Of Ethics Concerns:**

No ethics concerns.

---

> ### Author Response · Authors · 2023-11-17
> **Discussion on track**
>
> We appreciate your positive feedback, but are surprised you then reject it based on a mismatch of track. We do not think this is totally fair. Many similar papers to ours have appeared in the main track of important machine learning conferences in recent years, such as [1,2,3,4,5]. These papers all either present an environment suite, a benchmark set and/or an educational component, which our paper actually combines. (Note that our main focus is not benchmarking, but actually the environment suite and educational component.)
>
> We think such papers have a clear merit to the research community, even though they do not directly contribute a novel algorithm. The referenced algorithms are for example well cited, which shows the research community also values such efforts. Our paper specifically targets students entering the field, which should have a clear benefit, and it can also be used as an environment suite which disentangles challenges of RL. As such, we believe it could be very beneficial to the RL research community, and fits the ICLR main track as several other papers from recent years.
>
> We are eager to discuss this topic with you, since you otherwise seem very positive about our effort!
>
> 1. Osband, I., Doron, Y., Hessel, M., Aslanides, J., Sezener, E., Saraiva, A., ... & Van Hasselt, H. (2019, September). Behaviour Suite for Reinforcement Learning. In International Conference on Learning Representations.
> 2. Liang, E., Liaw, R., Nishihara, R., Moritz, P., Fox, R., Goldberg, K., ... & Stoica, I. (2018, July). RLlib: Abstractions for distributed reinforcement learning. In International conference on machine learning (pp. 3053-3062). PMLR.
> 3. Pleines, M., Pallasch, M., Zimmer, F., & Preuss, M. (2022, September). Memory Gym: Partially Observable Challenges to Memory-Based Agents. In The Eleventh International Conference on Learning Representations.
> 4. Terry, J., Black, B., Grammel, N., Jayakumar, M., Hari, A., Sullivan, R., ... & Ravi, P. (2021). Pettingzoo: Gym for multi-agent reinforcement learning. Advances in Neural Information Processing Systems, 34, 15032-15043.
> 5. Yu, T., Quillen, D., He, Z., Julian, R., Hausman, K., Finn, C., & Levine, S. (2020, May). Meta-world: A benchmark and evaluation for multi-task and meta reinforcement learning. In Conference on robot learning (pp. 1094-1100). PMLR.

---

### Official Review · Reviewer_kARL · 2023-11-03

**Soundness:** 2 fair
**Presentation:** 3 good
**Contribution:** 1 poor
**Rating:** 3
**Confidence:** 4

**Summary:**

This paper introduces a software environment for learning reinforcement learning. The software consists of a number of environments, explanations, and accompanying notebooks. The authors also conducted a human evaluation study and found that the majority of the participants find the proposed tool helpful.

**Strengths:**

This new RL environment for learning sounds like an exciting addition to the existing learning materials to help beginners to get into this field. The selection of problems and the sequence of presenting them seem well considered. The notebooks are detailed and easy to read. I think this will be a good educational resource.

**Weaknesses:**

The main contribution of this paper seems to be the creation of a new tool, software package, environment, and notebooks for learning RL. I am not sure how such contribution suits the themes at ICLR. While such new learning materials are welcomed, it lacks scientific rigor I am expecting from an ICLR contribution. In particular, the authors claim that the proposed environment is simpler and easier for people to learn RL. While the design of the proposed environment is supported by several reference literature articles, there is no comparison studies to demonstrate that the proposed environment is simpler, easier to learn, and overall better than other existing RL learning materials.

In addition, although the evaluation results show that EduGym improves learning outcomes, this is measured by self-reported survey. There is no other evaluative methods to assess learning, nor any kind of comparison (or A/B testing) to existing tools to benchmark the effectiveness of EduGym, without which it is difficult to understand the educational utility of the proposed environment.

To summarize, 1) this paper in my opinion is a good educational contribution rather than a technical contribution, which seems to me a unsuitable contribution for ICLR; 2) the paper lacks scientific rigor in terms of evaluating the utility of EduGym.

**Questions:**

I don't have questions for the authors.

---

> ### Author Response · Authors · 2023-11-17
> **Discussion on contribution and evaluation**
>
> You indicate our paper has an educational rather than technical contribution, which does not suit ICLR. We believe this criticism is not fair, when compared to the RL papers that have been published at major ML conferences in recent years. Our work contains an environment suite, of which several have appeared in big ML conferences in recent years [2,3], and can also be leveraged as a benchmark suite, which has also previously appeared at ICLR [1]. The same applies to structured codebases at the intersection of research and education [4]. All these papers got cited quite a lot, and have their impact on the community by making novel research feasible. Our additional educational focus should only add to this, since entering the research field can be challenging. In short, we believe our paper has merit to the research community, even though it does not present a new algorithm.
>
> Your second critique is that our work is not properly empirically validated. Of course, empirical validation is important, but we actually did include this evaluation. We asked a range of bachelor, master and PhD students with experience in RL whether they valued EduGym, which most of them strongly agreed with. These students already know a bit of RL, and can therefore judge whether our resources help them compared to what they used so far. Note that nearly all other teaching resources (books, public codebases) available are just published, and never evaluated with A/B testing. We therefore believe this criticism is also not completely fair, since we 1) included a useful practical evaluation and 2) already go beyond the practical evaluation of other educational resources.
>
> We are curious to hear your opinion on the above topics.
>
> 1. Osband, I., Doron, Y., Hessel, M., Aslanides, J., Sezener, E., Saraiva, A., ... & Van Hasselt, H. (2019, September). Behaviour Suite for Reinforcement Learning. In International Conference on Learning Representations.
> 2. Terry, J., Black, B., Grammel, N., Jayakumar, M., Hari, A., Sullivan, R., ... & Ravi, P. (2021). Pettingzoo: Gym for multi-agent reinforcement learning. Advances in Neural Information Processing Systems, 34, 15032-15043.
> 3. Pleines, M., Pallasch, M., Zimmer, F., & Preuss, M. (2022, September). Memory Gym: Partially Observable Challenges to Memory-Based Agents. In The Eleventh International Conference on Learning Representations.
> 4. Liang, E., Liaw, R., Nishihara, R., Moritz, P., Fox, R., Goldberg, K., ... & Stoica, I. (2018, July). RLlib: Abstractions for distributed reinforcement learning. In International conference on machine learning (pp. 3053-3062). PMLR.

---

### Meta-Review · Area_Chair_3KqZ · 2023-12-06

**Metareview:**

The paper introduces EduGym, a suite of educational RL environments to teach different concepts to students. EduGym provides interactive notebooks for students to learn and experiment with RL concepts in these environments. The reviewers appreciated the effort in developing open-source educational material for RL. However, the reviewers pointed out several weaknesses in the paper and raised concerns related to the limited technical contributions of the work. We want to thank the authors for their detailed responses. Based on the raised concerns and follow-up discussions, unfortunately, the final decision is a rejection. Nevertheless, the reviewers have provided detailed feedback and we encourage the authors to incorporate this feedback in their future work.

**Justification For Why Not Higher Score:**

The reviewers pointed out several weaknesses in the paper and raised concerns related to the limited technical contributions of the work. A majority of the reviewers think that the work is not suitable for publication at the ICLR conference.

**Justification For Why Not Lower Score:**

N/A

---

### Decision · Program_Chairs · 2024-01-16

Reject